# Mechanisms Analysis for Fatal Accident Types Caused by Multiple Processes in the Workplace: Based on Accident Case in South Korea

**DOI:** 10.3390/ijerph191811430

**Published:** 2022-09-11

**Authors:** Jeong-Hun Won, Hyeon-Ji Jeong, WonSeok Kim, Seungjun Kim, Sung-Yong Kang, Jong Moon Hwang

**Affiliations:** 1Department of Safety Engineering, Chungbuk National University, Cheongju 28644, Korea; 2Department of Big Data, Chungbuk National University, Cheongju 28644, Korea; 3Department of Disaster Prevention Engineering, Chungbuk National University, Cheongju 28644, Korea; 4Public Institution Assessment Bureau, Korea Occupational Safety and Health Agency, Ulsan 44429, Korea; 5School of Civil, Environmental and Architectural Engineering, Korea University, Seoul 02841, Korea; 6Occupational Safety and Health Research Institute, Korea Occupational Safety and Health Agency, Ulsan 44429, Korea

**Keywords:** accident mechanism, multiple processes, contractor, role and responsibility

## Abstract

This study aimed to develop the accident mechanism of fatal accidents taking place in multiple processes at the workplace. Multiple processes were defined as the existence of multiple work types and work processes in the same workspace. Recently, various processes are frequently conducted simultaneously in one workplace with the participation of several companies, and more workers are increasingly experiencing industrial accidents while working in multiple processes. To prevent accidents in the multiple processes caused by the sameness of work period and space, the accident process was investigated by analyzing the investigation reports on serious industrial accidents in South Korea, and then the accident mechanism model was developed. By utilizing the developed model, the major safety measures to be taken by the contractor for each of the 8 types of accidents caused by the multiple processes were drawn. The roles and responsibilities of the contractor to be implemented in order to prevent accident occurrence in multiple processes were proposed through the accident mechanism of each type of fatal accident. It is expected that the accidents taking place in the multiple processes can be prevented with the drawn results.

## 1. Introduction 

Recently, due to the enlargement, complication, and high-tech modernization of industries, various processes are frequently conducted simultaneously in one workplace, and several relevant contractors are frequently working simultaneously as a result of the generalization of contracted business. In such a case, interference in working space occurs in the work site due to the simultaneous processes, and as a result, the safety of the work decreases [1,2]. When at least two activities are executed within the same working space, workspace interference can occur. The problem of workspace interference has been considered through the overlapping period of activities; hence, many studies have been focused on solving workspace interference through scheduling activities [3]. In particular, workspace interference causes different challenges in a construction site, such as safety and productivity issues [4,5]. The project manager should consider the movability of the workspace and the criticality of an activity and resolve the problem of the workspace interference by an activity execution plan [6].

Figure 1 shows a comparison of the number of deaths in South Korea (henceforth, Korea) due to working and the number of deaths in the multiple processes, based on the preceding research of the authors [7]. Between 2016 and 2020, the number of deaths in the multiple processes amounted to 426 people, which is 7~14% of the total number of deaths by accidents each year. While the number of deaths by accidents due to working has been decreasing gradually, the number of deaths by accidents in the multiple processes has been increasing rapidly since 2017. In particular, the number of deaths due to the multiple processes increased in 2020 because 38 people died in a fire accident at the construction site of a new logistics center, called the Icheon Fire accident. The cause of the accident was an oil vapor explosion caused by simultaneous urethane foam work and metal cutting work, and it was classified as being due to multiple processes. As accidents are increasing in multiple processes, more efforts are being made to prevent the accidents by strengthening the contractor’s duty for safety and health management.

Japan’s Ministry of Health, Labor and Welfare implemented measures to prevent accidents occurring from working in the same place by appointing a safety and hygiene manager, and to maintain a general management system by appointing a specific original contractor in the construction and shipbuilding industries [8,9]. In Norway, it is required to appoint a general manager to take responsibility for the management of common safety in the common activities conducted simultaneously in the same place and in the common work site [10]. The Construction Design and Management Regulations in the UK (CDM 2015) assign responsibility to the original designer to plan and manage all of the stages of construction work to adjust matters related to health and safety, and if works are done simultaneously or consecutively, the designer has to consider the plan in the construction stage and the details of health and safety [11]. In Germany, an accident that has occurred by working in the same workplace as various subcontractors should be controlled [12,13].

Similar to other countries such as Japan, Norway, the UK, and Germany, Korea, too, is making efforts to prevent accidents caused by multiple processes with various systems. In Korea, accurate regulations on the prevention of accidents caused by multiple processes have recently been stipulated in the law based on cases in foreign countries. However, it is not based on research such as revealing the mechanism, as suggested in this study. As a first attempt to reduce accidents caused by multiple processes, the Korean government introduced the Safety and Health Coordinator System in 2017 for construction work sites and imposed the obligation to place a safety and health coordinator at the construction work site in order to prevent the industrial accidents that may occur due to the multiple processes conducted in the same place, for the contractor of a construction work where it is necessary to place two or more orders separately. The safety and health coordinator should grasp the details of the multiple processes of separately ordered works and the risk of the multiple processes, and should coordinate the work period and safety and health measures in order to prevent industrial accidents. 

However, the Safety and Health Coordinator System in Korea is limited to construction works with the scale of 5 billion won or more, and there is a problem that the responsibility for safety is unclear at the time of occurrence of an accident due to the multiple processes and the multistage subcontract structure in most industries. Therefore, in Korea, the responsibility of the contractor for safety and health was strengthened by expanding the range of the places where the contractor should take safety and health measures, from 22 dangerous places with the possibility of collapse, fall, etc., to the dangerous places controlled and managed by the contractor, by having revisions in the major points of the Occupational Safety and Health Act in 2019. In addition, Article 64 of the Occupational Safety and Health Act was revised on 29 April 2021. As shown in Table 1, No. 7 and No. 8 were added in Article 64 to make it obligatory for the contractor to coordinate the work period, contents, etc., of the work of relevant contractors, in order to minimize the risk of the multiple processes [14].

Workspace interference by multiple processes increases the potential safety and health risk. If workspace interference is not properly managed, accidents are still occurring. In construction, a more proactive method is being studied and applied to prevent the accident by workspace interference before construction. The building information method (BIM) is used as prevalent method to improve safety and health management in the construction industry [15]. Using BIM, potential safety hazards can be automatically identified and corresponding prevention methods can be applied from the design and planning stage before construction [16,17]. A BIM-based process can solve the workspace interference problem. In addition, it can monitor the workers’ unsafe behavior on site and enhance the current approach of the safety and health plan [18,19].

Though the responsibility of the contractor to take safety and health measures was strengthened and a more proactive active method was developed, accidents by workspace interference during multiple processes are still occurring. The safety measures of the contractor for multiple processes are still prescribed comprehensively and do not reflect the specific characteristics of the accidents. As it is necessary to consider the characteristics of the process of occurrence with the effect of the accident for prevention, it is important to understand the accident mechanism according to the time flow of the accidents [20]. In addition, as pointed out in the authors’ previous research [7], the existing accident mechanism does not correctly explain the accident process by multiple processes. Therefore, this study analyzed the investigation reports of fatal accidents caused by the multiple processes in order to discover a blind spot of safety and health caused by the multiple processes and to understand the characteristics of accidents, and it categorized the accident types to develop the accident mechanism.

## 2. Definition of Multiple Processes and Drawing of Accident Types

Four years’ worth of accident reports related to fatal accidents in the entire industries of Korea from 2016 to 2020 were examined based on the official accident data of the Ministry of Employment and Labor and the National Statistics Office, along with the report of the Korea Occupational Safety and Health Agency (KOSHA) on fatal accidents. KOSHA, which was established in 1987, is not only an organization standing at the front line to protect workers’ lives but also the only public organization for industrial accident prevention in Korea by conducting industrial accident prevention projects including R&D on industrial accident prevention techniques and providing technical guidance on occupational safety and health. The collected information on the accidents amounted to 4,641 cases in total. Among them, on the basis of the reports of accidents classified as accidents caused by the multiple processes, the reason for the accident, the process of accident occurrence, the original cause material, the cause of accident occurrence, the measures to prevent accidents, etc., were analyzed in order to draw the types of accidents caused by the multiple processes.

### 2.1. Definition of Multiple Processes in the Same Workplace

Multiple processes were defined as the existence of multiple work types and work processes in the same workspace through the analysis of process of accident occurrence in the investigation report of the accidents following the preceding research of the authors [7]. For the construction accident analysis model, many models have been developed such as the ConAC (Construction Accident Causation) framework, Bow-Tie, FTA (Fault Tree Analysis), and STAMP (System Theoretic Accident Model and Process) [20,21,22,23,24,25,26,27,28]. However, as pointed out in the authors’ previous research [7], the existing model is not correct for explaining the accident process by multiple process. This research used objective criteria for judgment of accident occurrence by the multiple processes in the same workplace, as presented in Table 2 below [7]. Based on the criteria, the process for determining the accident by multiple processes was drawn. The procedure for determining the accident by multiple processes is shown in Figure 2, and the notation for each decision criterion is shown in Table 2.

The procedure for determining the accident by multiple processes is divided into A, B, and C stages. A three-stage approach was proposed in the author’s previous study [7]. Since it may be difficult to generalize the path in multiple processes, the common criteria stakeholders, process, time, and space, were established as shown in Table 2 and Figure 2 to determine the accident by the multiple process. First, the A Stage is the stage of analyzing the relations of stakeholders. In this stage, an accident is classified into an accident between the contractor and the relevant contractor and an accident between the relevant contractors, aside from an independent accident of the contractor not prescribed in Article 64 of the Occupational Safety and Health Act. The B Stage is the stage of analyzing the work process. The cases were classified into an accident which occurred by the interference of two different processes and an accident which occurred by the different operations within one process. An example of a case of different processes conducted simultaneously is the operation of a machine while going through the maintenance. Lastly, the C Stage is the stage of analyzing the effect of the workspace and working time. The accidents occurring in the same working time and the accidents occurring in the same workspaces were classified, aside from the accidents which occurred during the independent operation of workers. Most accidents by the multiple processes occurred when both workspace and working time interfered; however, in other cases such as fire, explosion, and suffocation, some accidents occurred by the influence of residual toxic gas, etc., on the successive operation after the preceding operation conducted in the same workspace, even if the working time did not overlap.

### 2.2. Types of Accidents Caused by Multiple Processes

The accidents caused by the multiple processes were first classified into the construction industry, manufacturing industry, and other industries, and then classified into the 8 accident types that may be caused by the multiple processes, after analyzing the investigation report of fatal accidents. The classified accident types are ‘caught in equipment or machinery’, ‘collision’, ‘struck by the object (falling or flying)’, ‘fall’, ‘fire or explosion’, ‘crushed by machinery or equipment’, ‘collapse’, and ‘etc. (suffocation, poisoning, etc.)’, and the risks and characteristics of each accident type are shown in Table 3 below. The classification of the accident type is basically followed the general accident classification in Korea, and it is modified in detail by analyzing accident cases. The fatalities according to the considered accident type is tabulated in Table 4.

## 3. Analysis of the Process of Accident Occurrence by Multiple Processes

Among the fatal accidents classified as caused by the multiple processes in the analysis of investigation reports on serious industrial accidents, the process of accident occurrence of 61 cases with relatively detailed descriptions of the cause of accident were analyzed regarding the workspace and working time, according to the accident types drawn in Section 2.2.

### 3.1. Cases of Risk in Fire and Explosion

The results of analysis of the process of accident occurrence by the multiple processes in the cases of fire and explosion through the investigation reports on serious industrial accidents are in Table 5 below. In the analyzed cases, there are two types of occurrence of fire and explosion. The first type is a fire breaking out when welding in a space with inflammable gas or insulation material when different works are conducted in the same workspace. As welding and painting work, etc., are conducted simultaneously, a fire breaks out by the welding spark shooting up to inflammable gas or insulation material, which is vulnerable to fire. The second type is a fire or explosion breaking out in the successive work due to the preceding work when both are conducted consecutively in the same place. For example, when inflammable gas is accumulated from the preceding work, the workers in the successive work are not informed of the information and proceed with welding, and a fire breaks out as a result. In this case, the contractor must coordinate in advance to prevent the cause of fire or explosion caused by being conducted simultaneously with another work. If it is impossible to coordinate, it is necessary to check the safety measures of fire detector, etc., and to check and notify the risk of fire and explosion in the workspace of the successive work.

### 3.2. Cases of Being Caught in Equipment or Machinery

It was analyzed that most of the fatal accidents that are caused by being caught in machinery, equipment, etc., occur when two or more works are conducted simultaneously in the same place (Table 6). The accidents occurred when the machine operated abruptly or another worker operated the machine while the maintenance work was being done, without turning off the power of the machine, or when another worker operated the machine due to miscommunication. Thus, the contractor must check a protection device when using the machines and equipment and check the safety measures of the procedure in operating the machines and equipment in order to prevent accidents that can be caused by the abrupt operation of the machines and equipment during work. The results of analysis of the risk of multiple processes in being caught by the machinery, equipment, etc., through the investigation reports on serious industrial accidents are in Table 6 below.

### 3.3. Cases of Collision with Vehicles for Unloading and Transporting, Construction Machines, etc.

The accidents classified as collision with vehicles for unloading and transporting, construction machines, etc., by the multiple processes occur when two or more works are conducted in the same space and at the same time, and it was analyzed that most of the accidents occurred when the construction machine collided with another worker during the transport, as the operator of the machine was not aware of the nearby worker. The contractor must put the priority on adjusting the working time, and if it is impossible to coordinate the working time, the contractor must control the entrance of workers into the working area around the construction machine and allocate a signal man (Table 7).

### 3.4. Cases of Risk in Fall Accidents

The cases of fall accidents by the multiple processes are divided into the accidents occurring in the same space and time and the accidents in a different time in the same space due to the different working times. For example, in an accident where a worker fell and died while working on a workbench installed by another subcontractor, the result of the preceding work affected the successive work. Additionally, in a case in which the chain of a tower crane was fractured and a worker in another workspace was struck and fell to his death, this showed that an accident which occurred in one work may affect a worker doing another work in the same place and result in his death by falling. The contractor needs to separate the workspace by considering the scope of impact of the different works (Table 8).

### 3.5. Cases of Being Struck by the Object Falling

Mostly, struck accidents by a falling or flying object by multiple processes occur by the work areas located above and below, but some fatal accidents occur by being struck by an object flying from the adjacent upper space. Thus, the contractor needs to take measures to prevent the falling object from the adjacent upper workspace. The results of analysis of the risk of multiple processes in the cases of workers being struck by a falling or flying object through the investigation reports on serious industrial accidents are shown in Table 9 below.

### 3.6. Cases of Risk in Being Crushed by Machinery, Equipment, or Payload

Crushing accidents by machinery, equipment, or payload occur when two or more works are conducted in the same workspace and working time. In order to prevent accidents by the multiple processes, the contractor needs to implement the safety measures preventing workers being crushed by machinery, equipment, or payload and adjust the working time or workspace. The results of analysis of the risk of multiple processes in the cases of workers being crushed by the overturning of machinery, equipment, or payload through the investigation reports on serious industrial accidents are shown in Table 10 below. 

### 3.7. Cases of Risk in Collapse of Payload such as Soil and Sand, Structure, etc.

The accidents of collapse of the payload such as soil and sand, etc., by the multiple processes occur when two or more works are conducted simultaneously or when the preceding work affects the successive work. For example, an accident occurred due to the collapse of the payload during maintenance work when not being aware of the payload in the inside, due to the miscommunication between the contractor and the subcontractor. In addition, a worker working at the lower area was buried due to the collapse of piled soil and sand during the successive work. The results of analysis of the risk of the multiple processes in the cases of collapse of the payload such as soil and sand, structure, etc., through the investigation reports on serious industrial accidents are shown in Table 11 below.

### 3.8. Other Risks (Suffocation, Poisoning by Toxic Gas, etc.)

It was shown in the accident investigation report that the risk of suffocation and poisoning by toxic gas occurred in various types of multiple processes (Table 12). This can be classified into the accidents occurring in the same working time and workspace; the accidents of being exposed to toxic gas by the multiple processes at the same time in separated work spaces; and finally, the cases in which the toxic gas accumulated in the preceding work affects the successive work. It is necessary for the contractor to grasp the hazardous works that can affect the workspace before working and allow the successive work to be conducted only after measuring the concentration of toxic gas in the workspace. In addition, it is necessary to take safety measures such as checking and blocking the path of toxic gas connected to another workspace and adjusting the working time.

## 4. Development of Occurring Mechanism of Each Type of Fatal Accident by Multiple Processes

It is decided that fatal accidents occurring by multiple processes are caused by the chain reaction of various factors of accidents, as drawn in the analysis of risk in accident types. As mentioned earlier, the definition of multiple processes was drawn by conducting various work types and processes in the workspace. Therefore, in order to prevent accidents by multiple processes, it is important to grasp the safety measures that the contractor must implement for each accident type. This study examined first whether several works are conducted in the same workspace and working time and then developed the model of the accident occurring mechanism to judge the occurrence of accidents resulting from the safety measures by the contractor for each accident type (AMMP model).

The model of the accident occurring mechanism was developed on the basis of Event Tree Analysis (ETA), which is an inductive system analysis technique that grasps the process of accident occurrence as a chain process of the accident factors and expresses the chain progress from the triggering event of the accident to the actual occurrence of the accident in the shape of a branch. After analyzing the preceding research on the accident occurring mechanism, the basic event was grasped first by analyzing the root cause of the accident before using ETA technique, then the influence of the root cause on the accident was grasped and applied to the mechanism [29,30,31]. As the most basic condition of multiple processes presented in this study was the overlapping of the workspace and working time, the basic event was drawn on the basis of the flow of accident occurrence according to the overlapping of workspace and working time and the safety measures by the contractor analyzed in Chapter 3, through the investigation reports on serious industrial accidents.

The overall occurring mechanism of the accidents caused by the multiple processes was drawn by the following procedures on the basis of the basic definition of multiple processes.
(1)Checking the overlapping workspace of the multiple processes(2)Checking the overlapping working time according to the overlapping workspace(3)Grasping the accident types according to the overlapping workspace and working time

The result of the analysis was “No Accident,” as no accidents occurred unless the works were conducted in the same place and at the same time, as part of the concept of multiple processes. The conditions for occurrence of each accident type are shown in Figure 3 below.

### 4.1. Occurring Mechanism of Fire and Explosion

It was analyzed that the fire and explosion caused by the multiple processes occur when several works are conducted at the same time in the same workspace and when works are conducted successively in the same workspace. The accident occurring mechanism according to the role of the contractor is presented in Figure 4 below. From the accident occurring mechanism, in order to prevent fire and explosion accidents, the contractor must adjust the working time or check the fire watchman and the safety measures, if two or more works are conducted simultaneously in the same space, and must remove the inflammable gas generated in the preceding work for the successive works if not conducted simultaneously. If the contractor does not perform the presented roles, fire and explosion may be caused by the multiple processes.

### 4.2. Occurring Mechanism of Being Caught in Machinery, Equipment, etc.

The accidents of being caught in machinery, equipment, etc., caused by the multiple processes occur when several works are conducted simultaneously in the same workspace, and the accident occurring mechanism according to the role of the contractor is in Figure 5 below. If two or more works are conducted simultaneously in the same space, the contractor must adjust the working time or operate the machine after notifying other workers of the information of completion in the work and must check the safety measures in order to prevent accidents caused by the multiple processes. If the contractor does not perform the presented roles, the accidents of being caught in machinery, equipment, etc., may be caused by the multiple processes.

### 4.3. Occurring Mechanism of Collision with Construction Machines, etc.

It was analyzed that the accidents caused by collision with construction machines, etc., in multiple processes occur when several works are conducted simultaneously in the same workspace, and the accident occurring mechanism according to the role of the contractor is shown in Figure 6 below. If two or more works are conducted simultaneously in the same space, the contractor must adjust the working time, and if it is impossible to adjust the working time, the contractor must adjust the workspace. If it is impossible to adjust the workspace and the working time, the contractor must assign a watchman for the work, after checking the safety measures. If the contractor does not take such measures, the accidents of a collision with machines, etc., may be caused by the multiple processes.

### 4.4. Occurring Mechanism of Fall Accidents

It was analyzed that fall accidents caused by multiple processes occur when several works are conducted simultaneously in the same workspace and when works are conducted successively in the same place, and the accident occurring mechanism according to the role of the contractor is shown in Figure 7 below. If two or more works are conducted simultaneously in the same space, the contractor must adjust the working time, and if it is impossible to adjust the working time, the contractor must adjust the workspace. If the works are not conducted at the same time, the contractor must check what the preceding work is and then check the machinery and equipment in order to prevent fall accidents due to the insufficient safety inspection of machines and equipment. If the contractor does not take such measures, fall accidents may be caused by the multiple processes.

### 4.5. Occurring Mechanism of Being Struck by an Object

It was judged that the accidents of being struck by an object caused by multiple processes occur when several works are conducted simultaneously in the same workspace and when works are conducted in the upper and lower areas at the same time, and the accident occurring mechanism according to the role of the contractor is shown in Figure 8 below. If two or more works are conducted simultaneously in the same space, the contractor must adjust the working time, and if it is impossible to adjust the working time, the contractor must adjust the workspace. When works are conducted in the upper and lower areas at the same time, the accidents may occur due to the falling of an object from the upper workspace, and thus, it is necessary to take the safety measures.

### 4.6. Occurring Mechanism of Being Crushed by Machinery, Equipment, Payload, etc.

Accidents of being crushed by machinery, equipment, and payload occur when several works are conducted simultaneously in the same workspace, and the accident occurring mechanism according to the role of the contractor is in Figure 9 below. If two or more works are conducted simultaneously in the same space, the contractor must adjust the working time, and if it is impossible to adjust the working time, the contractor must adjust the workspace. If it is impossible to adjust the workspace and the working time, the contractor must check the machinery and equipment and conduct work only after checking the safety measures. If the contractor does not take such measures, the accidents of workers being crushed may be caused by multiple processes.

### 4.7. Occurring Mechanism of Collapse of Soil and Sand and Payload

Accidents of collapse of soil and sand and payload by multiple processes occur when several works are conducted simultaneously in the same workspace and when works are conducted successively in the same workspace (Figure 10). If two or more works are conducted simultaneously in the same workspace, the contractor must adjust the working time, and if it is impossible to adjust the working time, the contractor must control the dangerous area, take the safety measures, and assign a watchman. In the case of successive works in the same workspace without overlapping working time, the successive work must be conducted only after receiving the information of the preceding work.

### 4.8. Occurring Mechanism of Suffocation and Poisoning by Toxic Gas

It was analyzed that the accidents of suffocation and poisoning by toxic gas by multiple processes occur when several works are conducted simultaneously in the same workspace and when works are conducted successively in the same workspace, as well as when works are conducted at the same time in different workspaces, and the accident occurring mechanism according to the role of the contractor is shown in Figure 11 below. If two or more works are conducted simultaneously in the same space, the contractor must adjust the working time, and if it is impossible to adjust the working time, the contractor must control the dangerous area, take the safety measures, and assign a watchman. In the case of successive works in the same workspace without overlapping working time, the successive work must be conducted after measuring the concentration of toxic gas inside the workspace after the completion of preceding work. Additionally, if the work that discharges toxic gas in another workspace is conducted simultaneously, it is necessary to check and control the toxic gas intake passage of the other workspace.

## 5. Conclusions

This study analyzed the risk of each accident type and developed the accident occurring mechanism for industrial accidents by the multiple processes that have been increasing recently. By analyzing the fatal accident reports on the basis of the definition of multiple processes and the accident types drawn in the preceding research, the characteristics of industrial accidents by multiple processes was investigated.
(1)According to the developed accident occurring mechanism, the accidents of ‘being caught in equipment or machinery’, ‘collision with vehicles for unloading and transporting and construction machines’, and ‘falling and being crushed by machinery or equipment’ occur in the same workspace and working time, while ‘fire and explosion’, ‘fall accidents’, and ‘collapse of soil and sand and payload’ occur in the same workspace regardless of the working time. In addition, it was found that ‘being struck by an object’ occurs in the condition of works being conducted at the same time rather than being conducted in the same space, and ‘suffocation and poisoning’ occur in the condition of works either being in the same workspace or being conducted at the same time.(2)It was found that the role of the contractor is important for the reduction of risk of industrial accidents caused by multiple processes. The common role of the contractor in every accident type is to adjust the working time or separate the workspace, and to check the safety measures before proceeding with the work. If the works are conducted successively in the same space, it is necessary to give the information of the preceding work and notify the completion of the preceding work to the successive work, and to clean the environment of the workplace.(3)The major safety measures to be taken by the contractor for each of the 8 types of accidents caused by multiple processes were drawn as follows. If there is a risk of ‘fire and explosion’, the contractor must check if inflammable gas was generated in the preceding work conducted in the same place, and if there is inflammable gas, the contractor has to make sure that it has been removed. If the works are conducted at the same time, the working time must be adjusted. If there is a risk of ‘being caught in machinery, equipment, etc.’, the working time must be adjusted. If it is impossible to adjust the working time, the contractor must instruct the facility operator to share the information on the starting and end times of the works to the other workers, take the safety measures, and assign a watchman. If there is a risk of ‘collision with construction machine’ and ‘being crushed by machinery or equipment’, the contractor must adjust the working time and workspace and must check the safety of machinery and equipment. In the case of ‘fall accidents’ and ‘collapse of soil and sand and payload’, the contractor must check the preceding work and then inform the workers of the successive work of the safety information generated in the preceding work. In the case of ‘being struck by a flying or falling object’, the contractor must approve working only after checking the installation of safety devices such as a safety net against falling objects, as damage may be caused by an object falling or flying from an adjacent space rather than from the same space. Lastly, in the case of ‘suffocation and poisoning by toxic gas’, if it is impossible to adjust the working time, the contractor must check the concentration of toxic gas in the workspace after the completion of the preceding work and must take the safety measures in the toxic gas intake passage, in order to prevent the intake of toxic gas due to the adjacent work.

## Figures and Tables

**Figure 1 ijerph-19-11430-f001:**
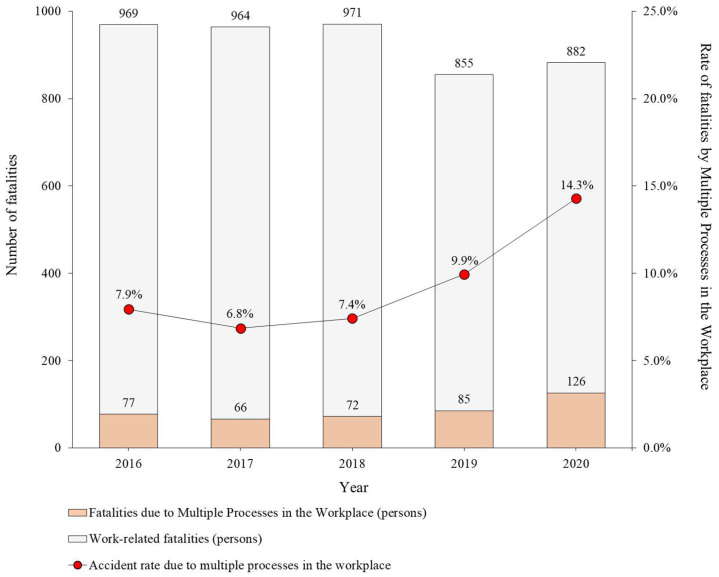
Percentage of fatalities caused by multiple processes in the workplace among work-related fatalities [7].

**Figure 2 ijerph-19-11430-f002:**
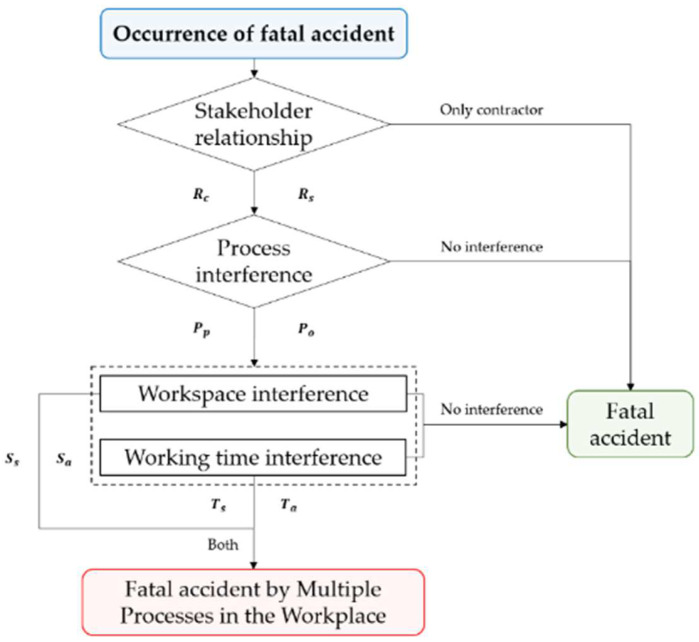
Procedures for determining accident by multiple processes in the workplace [7].

**Figure 3 ijerph-19-11430-f003:**
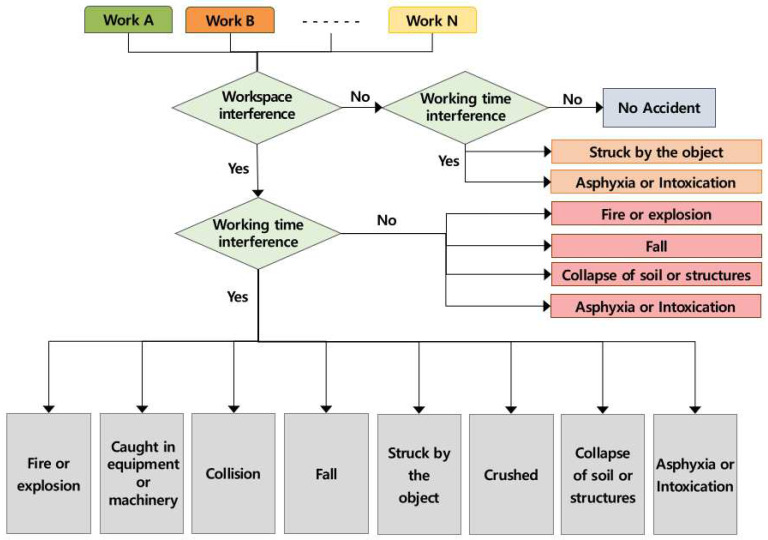
Accident occurrence mechanism by multiple processes in the workplace (AMMP model).

**Figure 4 ijerph-19-11430-f004:**
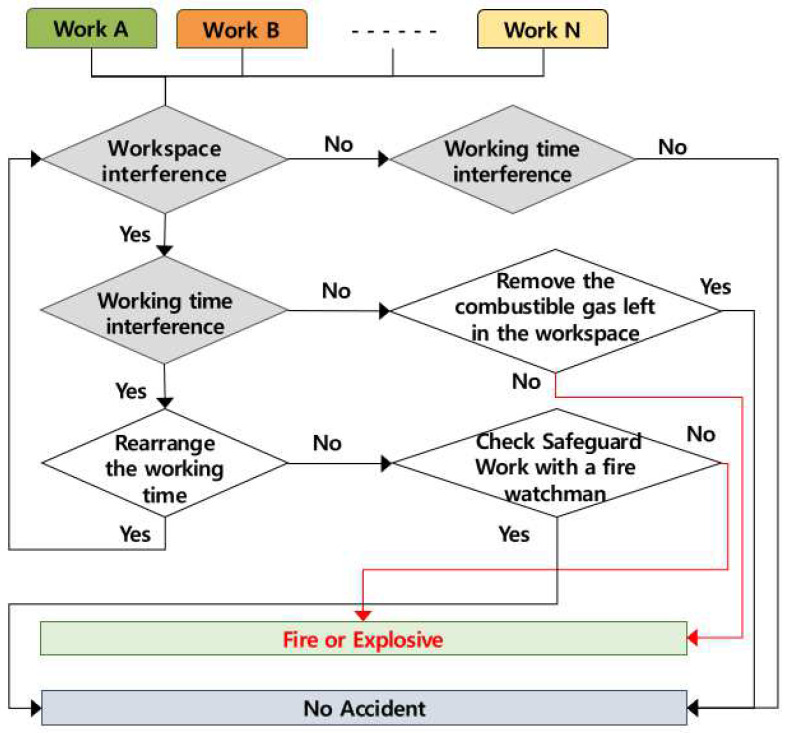
Mechanism of fire and explosion caused by multiple processes in the workplace (AMMP model—fire and explosion).

**Figure 5 ijerph-19-11430-f005:**
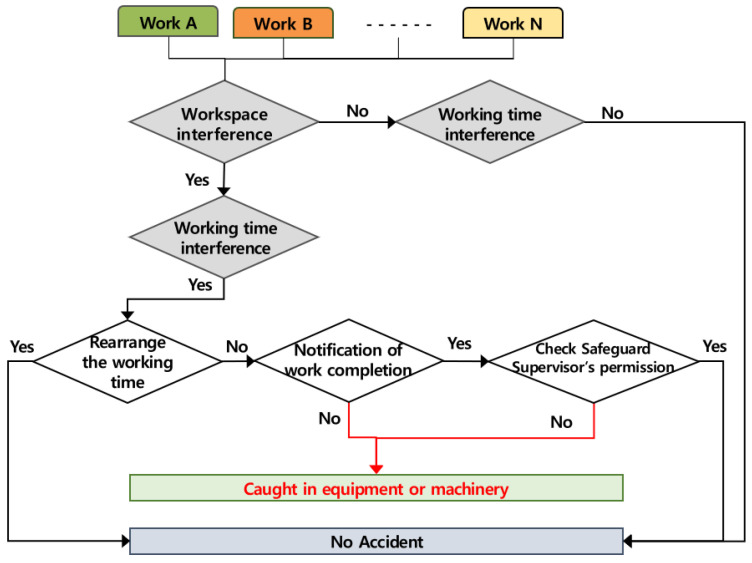
Mechanism of being caught in equipment or machinery by multiple processes in the workplace (AMMP model—caught).

**Figure 6 ijerph-19-11430-f006:**
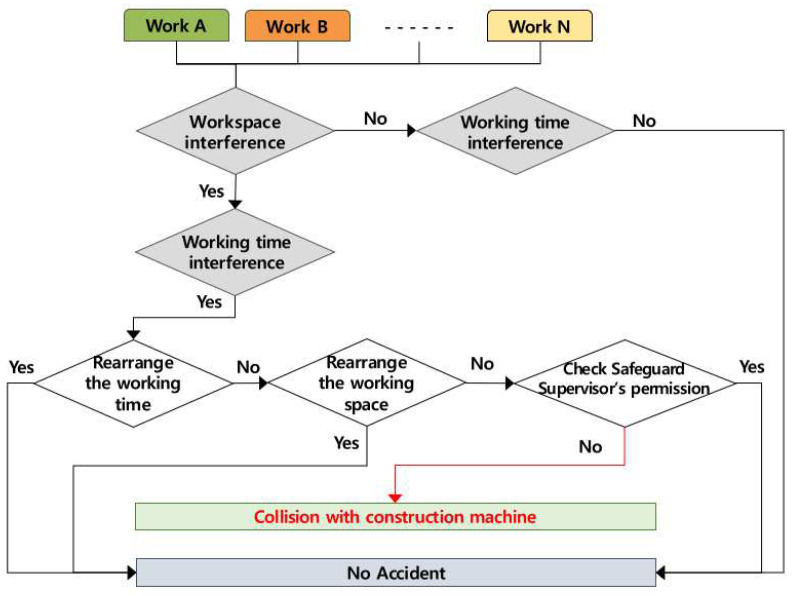
Mechanism of collision by multiple processes in the workplace (AMMP model for collision).

**Figure 7 ijerph-19-11430-f007:**
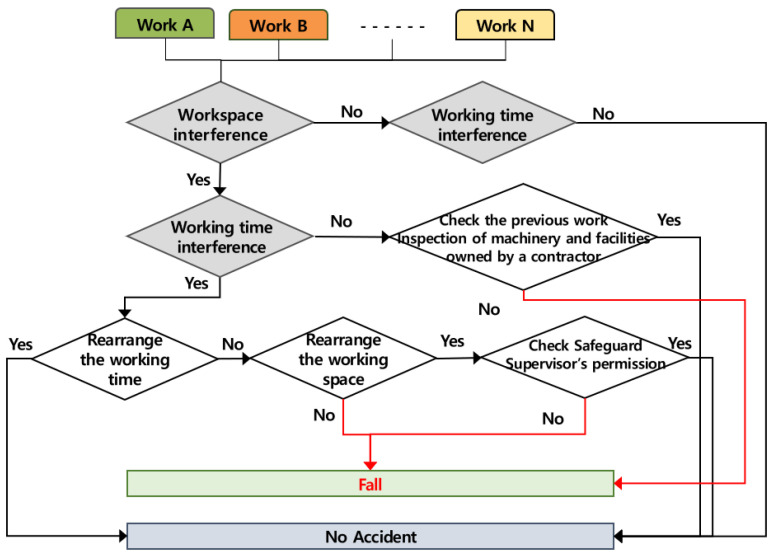
Mechanism of fall accident by multiple processes in the workplace (AMMP model for fall).

**Figure 8 ijerph-19-11430-f008:**
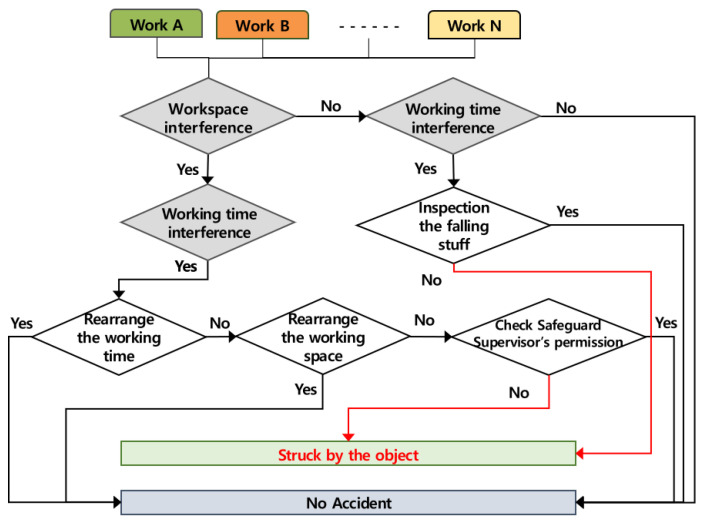
Mechanism of being struck by an object by multiple processes in the workplace (AMMP model for struck).

**Figure 9 ijerph-19-11430-f009:**
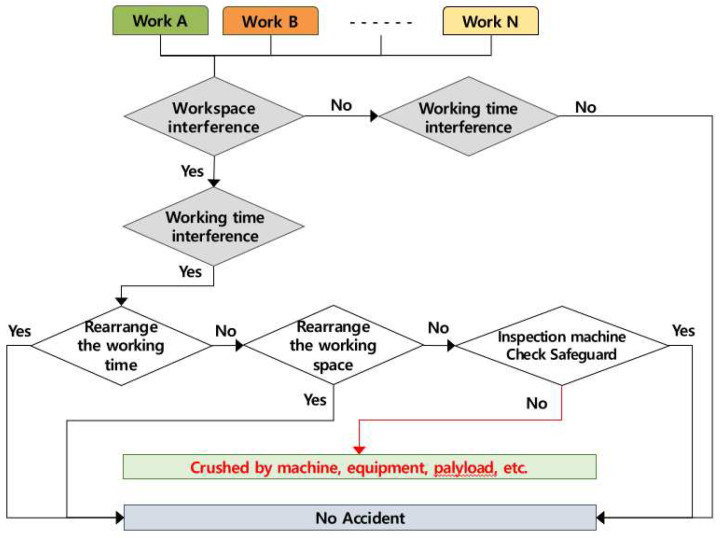
Mechanism of being crushed by multiple processes in the workplace (AMMP model for crushed).

**Figure 10 ijerph-19-11430-f010:**
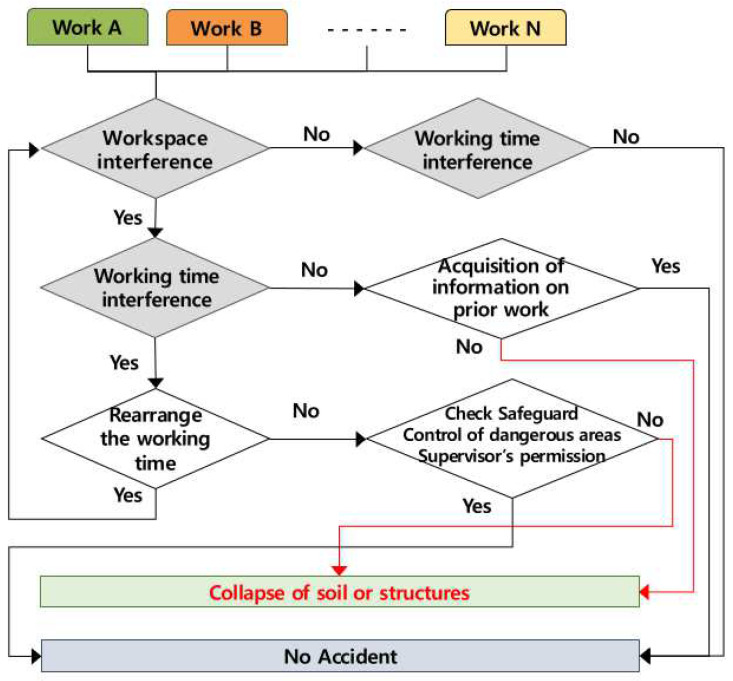
Mechanism of collapse by multiple processes in the workplace (AMMP model—collapse).

**Figure 11 ijerph-19-11430-f011:**
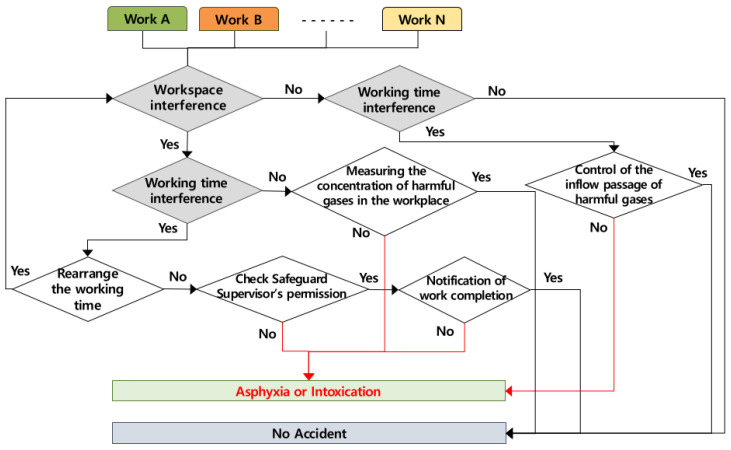
Mechanism of asphyxia or intoxication by multiple processes in the workplace (AMMP model for intoxication).

**Table 1 ijerph-19-11430-t001:** Occupational Safety and Health Act Article 64, Paragraph 18 (in Korea).

Article 64 Measures for Preventing Industrial Accidents in Contracting	
Paragraph 1	A contractor shall take the following measures when employees of a relevant contractor work at his or her place of business:
No. 1	Organizing and operating a council on safety and health consisting of a contractor and subcontractors
No. 2	A routine inspection of the workplace
No. 3	Supporting safety and health education that a relevant subcontractor provides to his or her employees pursuant to Article 29 (1) through (3), including providing the place and materials for such education
No. 4	Verifying the conduct of safety and health education that a relevant subcontractor provides to his or her employees pursuant to Article 29 (3)
No. 5	Operating a warning system, conducting evacuation drills, etc., in preparation for any of the following cases: (a) Where explosives are set for blasting at a work site (b) Where fire, explosion, collapse of soil, structures, etc., an earthquake, or any other accident occurs at a work site
No. 6	Providing places necessary to install facilities, etc., prescribed by Ordinance of the Ministry of Employment and Labor, such as sanitary facilities, or cooperating in the use of sanitary facilities installed by a contractor
No. 7	Confirming the work period and contents of the subcontractor, etc., safety measures, health measures, etc., in the work of a contractor and a subcontractor or subcontractors in the same workplace
No. 8	Adjusting the work period and contents of the subcontractors, etc., if there is a risk that a risk prescribed by Presidential Decree, such as fire or explosion, may occur due to the multiple processes in the workplace of related recipients, etc., according to the result of the confirmation of No. 7.

**Table 2 ijerph-19-11430-t002:** Decision process criteria and notation of multiple processes in the workplace.

Step	Classification	Decision Criteria	Notation
A	Stakeholder relationship	Accidents between a contractor and a subcontractor Accidents between subcontractors	$R_{c}$ $R_{s}$
B	Process interference	Accidents caused by other processes Accidents caused by two or more operations within a process	$P_{p}$ $P_{o}$
C	Workspace interference	Accidents occurring in the same workspaces Accidents occurring in adjacent workspaces	$S_{s}$ $S_{a}$
	Working time interference	Accidents occurring during the same working time Accidents caused by the previous process affecting continuous working time	$T_{s}$ $T_{a}$

**Table 3 ijerph-19-11430-t003:** Risks and characteristics of multiple processes in the workplace by accident types [7].

Accident Type	Risk of Fatal Accidents due to Multiple Processes in the Workplace by Industry	Interference Characteristics	
		Space	Time
Caught in equipment or machinery	When working with operating (powered on) equipment or machinery, there is a risk of interference with other operations performed in the same workplace at the same time.	Same workplace	Work at the same time
Collision	Risk arises when other operations are simultaneously interfered with within the working radius or movement range of the equipment, machinery, or means of transport used during the process.	Same workplace	Work at the same time
Struck by the object	When equipment, machinery, parts, or materials that have been displaced during work fly into or fall into another workspace where they are performed at the same time.	Upper and lower workplace or adjacent workplace	Work at the same time
Fire or explosion	Risk of interference when welding and cutting by operating equipment or machines at the same time in a workplace that uses inflammable substances or combustible gas.	Same workplace	Work at the same time
Crushed	Risk of interference if equipment, machinery, parts, materials, or vehicles fall over during work or at the same time invade the work area being carried out in the workplace.	Same workplace	Work at the same time
Fall	When equipment, machinery, parts, materials, or structures used for other work interfere with the working space of the worker and cause a fall.	Same workplace	Work at the same time or successive temporal relationships
Collapse	When a region or all of a part, material, or structure collapses and spills out and at the same time invades the workplace being carried out in the same place.	Same workplace	Work at the same time
Etc.	When performing other work in a space where oxygen deprivation or toxic gases that may cause suffocation or poisoning have accumulated or injected for work.	Same workplace	Work at the same time or successive temporal relationships

**Table 4 ijerph-19-11430-t004:** Fatalities based on the accident type caused by multiple processes in the workplace (person).

Year	2016	2017	2018	2019	2020
Caught in equipment or machinery	20	17	19	22	29
Collision	24	15	20	23	23
Struck by the object	11	15	10	18	17
Fire or explosion	9	3	2	3	36
Crushed	4	8	7	12	9
Fall	4	3	7	1	4
Collapse	2	3	2	2	7
Others	3	2	5	4	1
Total	77	66	72	85	126

**Table 5 ijerph-19-11430-t005:** Examples of fire and explosion accidents by multiple processes in the workplace.

Industrial Accident Occurrence Process	Same Space Work (○, x)	Same Time Work (○, x)	Safety and Health Measures of Contractor against Accidents Caused by Multiple Processes
Air leaked during blind patch bolt locking operation of By-Pass Filter Tank airtight test. Bolt locking work and compressed air injection work were carried out at the same time. →By-Pass Filter Tank explosion caused subcontractor worker’s death	○	○	Supervise to ensure that both hazardous operations do not occur at the same time (bolt locking, compressed air injection)
The snuff occurred during welding work by a daily employed weld worker belonging to a contractor. Fires in insulation due to the snuff→Many workers working on other jobs died.	○	○	If the welding work is performed in the same place or on the same floor, the welding work is finished and another operation is performed. If work needs to be done on another floor, check safeguard and the placement of fire watchman
Welding and painting work were carried out simultaneously. Welding sparks scattered in combustible gas generated from painting and exploded→Subcontractor worker’s death.	○	○	Adjust welding working time or separate it from other work. When welding and other work are performed at the same time, a supervisor is arranged to check the explosion hazard factors in advance and implement ventilation.
Combustible gas accumulated in the workplace due to another work performed before welding work→ Worker death due to explosion caused by welding work.	○	x	When welding and other work are performed at the same time, a supervisor is arranged to check the explosion hazard factors in advance and implement ventilation.

**Table 6 ijerph-19-11430-t006:** Cases for being caught in equipment or machinery by multiple processes in the workplace.

Industrial Accident Occurrence Process	Same Space Work (○, x)	Same Time Work (○, x)	Safety and Health Measures of Contractor against Accidents Caused by Multiple Processes
The subcontractor’s worker used the machine and equipment owned by the contractor to work → The subcontractor’s worker died by being caught in machinery and equipment while working.	○	○	The contractor must check the protective measures for machinery and equipment before the subcontractor performs the work. Working in pairs of two.
The contractor’s machinery and equipment suddenly operated while the subcontractor’s worker performed maintenance/inspection/repair/repair/cleaning of the machine and equipment → The subcontractor’s worker died from being caught in machinery and equipment.	○	○	Procedures should be established to ensure that machinery and equipment are not operated until work is completed. Notify when subcontractor’s work is complete.
Although the machine and equipment inspection work was not completed, the contractor’s driver decided the work had been completed and operated → The subcontractor’s worker died from being caught in machinery and equipment.	○	○	The communication of information on completion about inspection of machinery and equipment and the operating system should be confirmed, and machine inspection and operating time should be separated.
When worker had replaced the sensor in the upper part of the injection molding machine, another worker operated the overhead crane for transport → The contractor worker died from being caught in machinery and equipment.	○	○	If the worker does other work within the working range of the overhead crane, the crane work must be stopped.
The contractor’s industrial robot malfunctioned → Subcontractor’s worker died by hitting the industry robot.	○	○	Industrial robot must be inspected.

**Table 7 ijerph-19-11430-t007:** Cases of collision accidents by multiple processes in the workplace.

Industrial Accident Occurrence process	Same Space Work (○, x)	Same Time Work (○, x)	Safety and Health Measures of Contractor against Accidents Caused by Multiple Processes
Excavator was used to transport soil → A subcontractor worker performed a wire clean up after a sleeve work was hit by an excavator and died.	○	○	Working time must be adjusted. If it is impossible to adjust the working time, a signal man must be allocated and workers must be prohibited from entering within the working area of the excavator.
The subcontractor’s construction machinery was going backwards → A worker got caught and died during cleanup.	○	○	Working time and workplace must be adjusted. Must check signal man arrangement plan.
Workers carried out waste loading by operating excavators and dump trucks → A subcontractor worker while cleaning up residue was hit and died.	○	○	Since it is difficult to separate the working time and space, the signal system and the signal man arrangement plan should be checked before work.
Excavator overturned while moving on a slope → A worker died while working on a water pipe splicing.	○	○	Working time should be adjusted to stop other work during excavator moving work.

**Table 8 ijerph-19-11430-t008:** Cases of fall accidents by multiple processes in the workplace.

Industrial accident Occurrence Process	Same Space Work (○, x)	Same Time Work (○, x)	Safety and Health Measures of Contractor against Accidents Caused by Multiple Processes
Fracture of contractor’s machinery and equipment, failure of safety devices → Subcontractor worker fell to their death.	○	x	Since the contractor’s machinery and equipment used by the subcontractor are used in the same space, the contractor must inspect the machinery and equipment and confirm the safety certification before work.
The workbench installed by the contractor or other subcontractor broke off or fell off → The subcontractor worker who was working on the workbench fell and died.	○	x	Since the workbench installed by another subcontractor is used in the same space, it is necessary for the contractor to inspect the workbench safety before work.
Impacted by the material that fell out during the work of dismantling the Soil–Cement Wall → The signal man fell and died.	○	○	Examine the scope of impact of the work and check the plan, such as placement of signal numbers.
After dismantling the form, the form fell due to the falling off of the sling belt while moving → Another subcontractor worker on the scaffold fell due to the impact.	○	○	Temporarily separate the worker’s movement path from the work area during the lifting operation.

**Table 9 ijerph-19-11430-t009:** Cases of struck accidents by the object by multiple processes in the workplace.

Industrial accident Occurrence Process	Same Space Work (○, x)	Same Time Work (○, x)	Safety and Health Measures of Contractor against Accidents Caused by Multiple Process
Part of the subcontractor worker’s equipment fell off → Another subcontractor worker who was working in the lower part of the workspace was struck by the object and died.	○	○	Safety measures against equipment falling off should be checked.
Breakage of lifting mechanism, malfunction of machinery and equipment, fall of lifting object → Another subcontractor worker who was working in the lower part of the workspace was struck by the object and died.	○	○	Arrangements should be made to temporarily prohibit workers from working in the lifting path.
The object of the subcontractor worker who was working on the upper part fell → Another subcontractor worker who was working in the lower part of the workspace was struck by the object and died. (The two workplaces are not on the same floor, but are placed vertically.)	x	○	After checking the measures to prevent falling objects in the upper work area, work must be done.

**Table 10 ijerph-19-11430-t010:** Cases of crushed accidents by multiple processes in the workplace.

Industrial accident Occurrence Process	Same Space Work (○, x)	Same Time Work (○, x)	Safety and Health Measures of Contractor against Accidents Caused by Multiple Process
Moving material handling equipment collided with a loaded heavy object → The worker was crushed by a heavy load during inspection.	○	○	The time for inspection work must be adjusted while the heavy object is moving.
The heavy object piled up during the sculpture loading operation fell down → The worker who was working on the sculpture painting work was crushed.	○	○	Check work done within the impact range of heavy load and check safety measures.
The table foam being moved by the forklift collapsed → Workers working on the height adjustment of the form were crushed.	○	○	It is necessary to check the types of work performed in the moving work route in advance and check safety measures.
The crane fell down → The worker was crushed.	○	○	Machines and instruments must be inspected.

**Table 11 ijerph-19-11430-t011:** Cases of collapse accidents by multiple processes in the workplace.

Industrial accident Occurrence Process	Same Space Work (○, x)	Same Time Work (○, x)	Safety and Health Measures of Contractor against Accidents Caused by Multiple Process
During maintenance work, an error occurred in communication between the contractor and the subcontractor; and it was insufficient to identify the load inside the tank → The worker was crushed and died due to the collapse of a heavy object during the maintenance work.	○	x	The contractor must communicate and confirm accurate information about the maintenance work to the subcontractor.
During the demolition work, the slab could not endure the load of the excavator and collapsed → A worker during the watering operation was buried and died.	○	○	During demolition work, it is necessary to check the measures to separate workers within the range of influence.

**Table 12 ijerph-19-11430-t012:** Cases of asphyxia or intoxication accidents by multiple processes in the workplace.

Industrial Accident Occurrence Process	Same Space Work (○, x)	Same Time Work (○, x)	Safety and Health Measures of Contractor against Accidents Caused by Multiple Process
Gas remained in the pipe → A worker who entered the pipe for welding work died from oxygen starvation.	○	×	Before welding, the work must be permitted after measuring the gas generated in the preceding work.
Nitrogen flew in from machine operation while working inside the cooling tower → Worker died due to a lack of oxygen.	x	○	It is necessary to permit work after preventing the inflow of gases such as nitrogen into the path leading to the enclosure.
Operation of vacuum chamber incomplete → Death from asphyxia.	○	○	After confirming that the number of workers put into the work matches the number of workers who came out after completing the work, the operation measures should be confirmed to operate the equipment.
Sulfuric acid leaked when entering the repair work area → Death due to burns.	○	○	Safety and health information should be provided to relevant contractors beforehand and risk prevention measures should be installed.

## Data Availability

Data are contained within the article.

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
