# Peer review of "Mechanisms Analysis for Fatal Accident Types Caused by Multiple Processes in the Workplace: Based on Accident Case in South Korea"

_ijerph, 2022, doi:10.3390/ijerph191811430_

Round 1

Reviewer 1 Report

The manuscript seems to well structured.

1. The authors need to justify the reason for sudden increase in fatalities due to multiple processes in workplace in the year 2020. 

2. In third paragraph, the authors discussed about various countries implemented certain protocols to prevent accidents, Japan, UK, German etc., in what way it useful for this study? If so specify clearly the usage of referred articles. If possible summarize the literature survey and add at the Introduction.

3.  What is the significance of Table 1? If the sections of an Act is used or discussed make more clarity on it. It says only 7 & 8 are referred and 1-6 omitted, that is not the right way of tabulating the points.

4. The authors need to add more information on KOSHA, when it is established and what is the roles and responsibilities.    

5. Any previous reference for three steps classification, if so cite the reference. If it is authors idea or concept try to justify its method.

6. In Table 3, how the accident type is classified? Any reference?

7. How Work A, B, C, & D is derived? Any specific protocols used?

8. Any numerical data analysis for the figures 4 to 10? If so try to add the details, since it seems to generic and weak discussions.

9. Add few more latest references.

Author Response

Thank you so much for your comments and suggestions.

I greatly appreciate your efforts and helpful comments in reviewing our paper.

Please see the attached files for the response, and the revised or supplemented content is marked in red in the revised paper.

Reviewer 2 Report

This study attempted to develop the accident mechanism of fatal industrial accidents taking place in the multiple processes in South Korea’s workplaces. Based on the investigation reports on fatal industrial accidents (mainly in the construction and manufacturing industries), the accident process was analyzed and the accident mechanism model was developed. The contractor’s major safety measures for eight types of accidents caused by the multiple processes were also drawn utilizing the developed models. The roles and responsibilities of the contractor, in terms of preventing fatal accidents occurring in multiple processes, were also proposed.

This paper can be considered a follow-up study (part II of the research) of Reference #7 (part I). Figures 1 & 2 of this paper are identical to Figures 4 & 3 in Reference #7, respectively. Please correct the citation of Figure 2 to [7].

The research topic is an important and interesting one. In addition to the overall quality, the paper’s content, methodological application, knowledge commitment, and contribution have met the requirement of an academic journal. For these reasons, it is recommended that the paper be accepted.

Author Response

Thank you so much for your comments and suggestions.

I greatly appreciate your efforts and helpful comments in reviewing our paper.

Please see the attached file for the response.
